# Remimazolam-etomidate versus remimazolam-propofol for gastrointestinal endoscopy: A randomized controlled trial

**Baoyu Ma**, **Ning Zhang**, **Rong Huang**, **Shoushi Wang** *

Department of Anesthesiology and Perioperative Medicine, Qingdao Central Hospital, University of Health and Rehabilitation Sciences (Qingdao Central Medical Group), Qingdao, China

* wangshoushi1226@126.com

**Data availability statement:** All relevant data are within the article and its Supporting Information files.

**Funding:** The author(s) received no specific funding for this work.

## Abstract

### Objective

The optimal sedation strategy for gastrointestinal endoscopy remains debated. This study compared the efficacy and safety of remimazolam combined with etomidate versus propofol for procedural sedation during gastrointestinal endoscopy.

### Methods

This single-center, randomized controlled clinical trial was performed from March 2024 to April 2024. A total of 262 patients scheduled to undergo gastrointestinal endoscopy were randomly assigned to receive remimazolam-etomidate (RE) or remimazolam-propofol (RP). The primary outcome was the incidence of respiratory depression. Secondary outcomes included the results of sedation and recovery. The safety results mainly include the incidence of hypotension, bradycardia, tachycardia, painful injection and muscular tremor. Statistical analyses included t-tests, Mann-Whitney U tests, and $\chi^2$ tests for group comparisons, with subgroup analyses and multivariable logistic regression to assess the robustness of primary outcome.

### Results

Respiratory depression occurred in 20.0% (25/125) of RE patients versus 32.3% (40/124) of RP patients (OR=0.52; 95% CI=0.29–0.93; $p$=0.028). There was a statistically significant difference in the distribution of the number of airway interventions between the two groups ($p$=0.043), with 18 patients (14.5%) in the RP group requiring three airway interventions and only seven patients (5.6%) in the RE group. Hypoxemia occurred in three patients (2.4%) in the RE group and in five patients (4.0%) in the RP group. Hypotension was observed in 23.2% of patients sedated with RE versus 36.3% of patients sedated with RP ($p$=0.024).

**Competing interests:** The authors have declared that no competing interests exist.

## Conclusion

Remimazolam-etomidate demonstrated a superior safety profile, with reduced respiratory depression and hemodynamic instability compared to remimazolam-propofol, suggesting its potential as a safer alternative for gastrointestinal endoscopy sedation.

## Registration information

This trial was prospectively registered at the Chinese Clinical Trial Registry (ChiCTR2400085904) prior to patient enrollment.

---

## Introduction

Gastrointestinal endoscopy is widely recognized as the gold standard for diagnosing gastrointestinal tract diseases and is frequently employed as the primary investigative approach [1]. Procedural sedation is essential to alleviate patient discomfort during invasive endoscopic examinations [2]. Although procedural sedation is generally considered safe, adverse events remain unavoidable, and the optimal sedation regimen for endoscopy continues to be a subject of debate.

Propofol, etomidate, and midazolam combined with opioids are the most commonly used drugs globally for sedation during endoscopic procedures [3]. Among them, propofol is the most preferred for procedural sedation owing to its superior sedative properties. However, its drawbacks include injection pain, a narrow therapeutic index, and the potential for cardiovascular and respiratory depression, particularly when co-administered with opioids [4]. Etomidate exerts minimal inhibitory effects on sympathetic tone and myocardial function, resulting in fewer adverse cardiovascular effects. However, when used alone, etomidate may induce muscle tremors and rigidity, as well as an increased incidence of postoperative nausea and vomiting (PONV) [5]. Midazolam, renowned for its potent amnestic properties, is the most frequently utilized benzodiazepine. However, its long half-life and active metabolites compromise its controllability [6]. Remimazolam, a short-acting gamma-aminobutyric acid A receptor agonist, exhibits a brief half-life and has demonstrated safety and efficacy as an option for procedural sedation [7–8]. In clinical practice, remimazolam alone is frequently inadequate to manage the intense stimulation associated with gastroscope placement. Increasing the dose to achieve sufficient sedation depth elevates the risk of adverse events, including hypotension and respiratory depression [9]. Each sedative agent possesses distinct advantages and drawbacks, and the development of novel drugs remains challenging. Consequently, combining the strengths and weaknesses of existing drugs through compounding represents a rational approach. Accordingly, this study aims to assess the efficacy and safety of remimazolam in combination with either etomidate or propofol for procedural sedation during gastrointestinal endoscopy.

## Methods

### Study design

This single-center, randomized controlled trial was approved by the Medical Ethics Committee of Qingdao Central Hospital Medical Group: KY202404801. Trial was conducted in accordance with the Declaration of Helsinki and International Conference on Harmonisation of Good Clinical Practice. This trial was prospectively registered at the Chinese Clinical Trial Registry (ChiCTR2400085904) prior to patient enrollment. Patients were recruited from March 23, 2024, to April 30, 2024, the last patient's follow-up was also completed on April 30, 2024. Written informed consent was obtained from each patient, before the start of any protocol-specified procedures.

### Patients

The inclusion criteria for the trial were as follows: ASA I to III adults undergoing elective gastrointestinal endoscopy under procedural sedation. The exclusion criteria were patients with severe respiratory depression, acute or severe bronchial asthma, known or suspected gastrointestinal obstruction, known allergy to the drugs used in this study, severe hepatic and renal insufficiency, long-term use of opioids, long-term use of benzodiazepines, anticipation of difficult airway, adrenocortical insufficiency.

### Randomization and blinding

The researchers used the website (www.random.org) to generate the block group randomization scheme. All included participants were randomly assigned in a 1:1 ratio to receive either remimazolam combined with etomidate or propofol for procedural sedation. Groups were blinded to participants, anesthesiologists, endoscopists, and postoperative follow-up staff.

### Procedures

All participants were required to undergo a preoperative evaluation in the anesthesia clinic. Baseline information including age, gender, height, weight, ASA classification, Mallampti classification, history of smoking, history of alcohol consumption, history of surgery, comorbidities, and PRODIGY score [10] were recorded.

Upon admission to the operating room, the patient is asked about the duration of food and drink abstinence, monitoring equipment is connected to detect vital signs including $SpO_2$ (Peripheral Capillary Oxygen Saturation), NIBP (Non-Invasive Blood Pressure), electrocardiogram, and $P_{ET}CO_2$ (End-Tidal Carbon Dioxide). Adequate oxygenation with endoscopic mask under spontaneous respiration (8–10 L/min for 3–5 min). Nursing staff prepared the sedative medicines according to the randomization group. Propofol and etomidate were diluted to the same volume and labeled using special labels. Oxygen (6 L/min) was given using an endoscopic mask throughout the endoscopic maneuver until the patient was fully awake.

Analgesia was provided with sufentanil (5−10 mcg), and sedative medication was administered three minutes after sufentanil was given. Performing an intravenous injection of remimazolam (0.15 mg/kg) over one minute using microinjection pumping as recommended in the instructions. After the remimazolam injection was completed, the induction of anesthesia was accomplished with a slow injection of etomidate (0.1 mg/kg) or propofol (0.75 mg/kg). Gastrointestinal endoscopy was initiated when the patient was adequately sedated (Modified Observer's Assessment of Alertness/Sedation [MOAA/S] ≤ 3). Supplemental medication (remimazolam 2.5 mg) may be given after a one-minute interval from the end of the initial dose if sedation is not considered adequate or if gastroscopic entry is failed. A maximum of five supplemental doses were given within 15 minutes during sedation maintenance, otherwise sedation was judged to have failed. If the initial and supplemental doses were insufficient to obtain adequate sedation for endoscopy, remedial medication (propofol 0.75 mg/kg) was administered at the discretion of the anesthesiologist. All sedation procedures were performed under the supervision of board-certified anesthesiologists with specialized training in endoscopic sedation. This included

medication administration, physiological monitoring, and depth of sedation assessment. Sedation levels were assessed at three-minute intervals throughout the procedure, maintaining MOAA/S ≤ 3. Flumazenil (0.2–0.5 mg) was given after withdrawing from the colonoscopy, after which the level of sedation was assessed at one-minute intervals until the patient was fully awakened (three consecutive MOAA/S scores of 5).

## Outcomes

The primary outcome was the incidence of respiratory depression. Respiratory depression was defined as a respiratory rate <8 breaths/min and/or $SpO_2 < 90\%$ [10]. Respiratory rate is monitored by measuring $P_{ET}CO_2$, and any observed decrease in respiratory rate or $SpO_2$ prompts the anesthesiologist to verify the accuracy of these readings to confirm they remain within normal limits. Ineffective breathing due to tongue retroversion may occur, and if airway obstruction is alleviated through airway maneuvers, a respiratory rate exceeding the assessment threshold at that point is not classified as respiratory depression. However, the type and frequency of airway maneuvers must be recorded.

The other outcomes included the incidence of hypoxemia ($SpO_2 < 90\%$, >10s), number of airway interventions, number of endoscopes removed due to hypoxemia, success rate of sedation. The time of procedure (placement of gastroscope to withdrawal of colonoscopy), time of sedation (administration of analgesic medication to complete awakening), and time of awakening (administration of flumazenil to complete awakening) were recorded. The number of sedative medication refills, the dose of sedative medication administered, and the dose of analgesic medication administered were recorded. Mean Arterial Pressure (MAP), Heart Rate (HR), $SpO_2$ were recorded at different time points. Immediate postoperative nausea and vomiting scores were documented once patients were fully awake. Following the procedure, endoscopist satisfaction was assessed using a 10-point scale, with scores of 0–3 classified as unsatisfactory, 4–7 as relatively satisfactory, and 8–10 as satisfactory. On the first postoperative day, patient satisfaction was evaluated based on subjective self-reports (scored out of 10, with 0–3 deemed unsatisfactory, 4–7 relatively satisfactory, and 8–10 satisfactory), alongside records of nausea and vomiting.

Finally, the adverse effects during the procedure were noted, including hypotension (20% decrease in systolic blood pressure from baseline or MAP<60 mmHg), bradycardia (HR<60 beats/min, >10s), tachycardia (HR>100 beats/min, >10s), painful injection and muscular tremor.

## Statistical analysis

Based on the summary of previous studies and pre-test data, the incidence of respiratory depression during gastrointestinal endoscopy with remimazolam combined with propofol was taken as 30%. Thus, assuming that remimazolam in combination with etomidate can reduce the incidence of respiratory depression to 15%, we estimated a sample size of 118 patients in this study, which had 80% power to detect a significant difference level of 0.05. Considering a dropout rate of 5%, we enrolled 124 patients in each group.

Statistical analysis was performed using SPSS software (ver. 26.0; SPSS Inc., Chicago, IL, USA). Continuous variables are expressed as mean ± standard deviation (SD) or median (P25, P75). The Kolmogorov-Smirnov test was used to check the normal distribution of the data for continuous variables. Continuous variables that conformed to normal distribution were tested using the two independent samples t-test, and continuous variables that did not conform to normal distribution were tested using the Mann-Whitney U test. Categorical variables were expressed as number and frequency and analyzed using the χ2 test or Fisher's exact test. A $p < 0.05$ was considered to be statistically significant. A subgroup analysis was performed to find differences in the primary outcome according to the age, sex, BMI, ASA, comorbidities, sleep apnea history and PRODIGY score. Further, we performed sensitivity analyses using multivariable logistic regression for primary outcome. Covariates included age, sex, BMI, ASA, comorbidities, sleep apnea history and PRODIGY score, selected based on clinical relevance.

## Results

### Patient demographics and baseline characteristics

Fig 1 illustrates the patient enrollment process for this trial. Between March and April 2024, we screened 262 patients scheduled for gastrointestinal endoscopy. Of these, six were excluded for the following reasons: four were on long-term benzodiazepines for insomnia, one had hearing loss, and one exhibited poor communication skills. Consequently, 256 eligible patients were randomized to receive either RE sedation (n = 128) or RP sedation (n = 128). In the RE group, three patients did not complete the trial: one was found to have a pharyngeal mass during gastroscopy and was withdrawn due to bleeding risk, while two had their endoscopists opt to perform a colonoscopy first. In the RP group, three patients' endoscopists prioritized colonoscopy, and one patient underwent only gastroscopy due to inadequate bowel preparation. Ultimately, 125 patients in the RE group and 124 in the RP group completed both the study and follow-up.

Both groups were well balanced at baseline (Table 1), with a median age of 58 (47,65) and 55.5 years (43,61.75) in the RE and RP groups, respectively ($p = 0.125$). There was no statistically significant difference in the procedure time [32(27,38) vs. 31(26,35)], sedation time [40(35,48) vs. 40(35,44.75)], and awake time [4(4,5) vs. 4(4,4)] between the two groups. There was no statistical difference in the amount of remimazolam [23.4 ± 4.5 vs. 23.2 ± 4.1], sufentanil [5(5,5) vs. 5(5,5)], flumazenil [0.3(0.3,0.3) vs. 0.3(0.3,0.3)] and intraoperative fluids [500(500,500) vs. 500(500,500)] used between

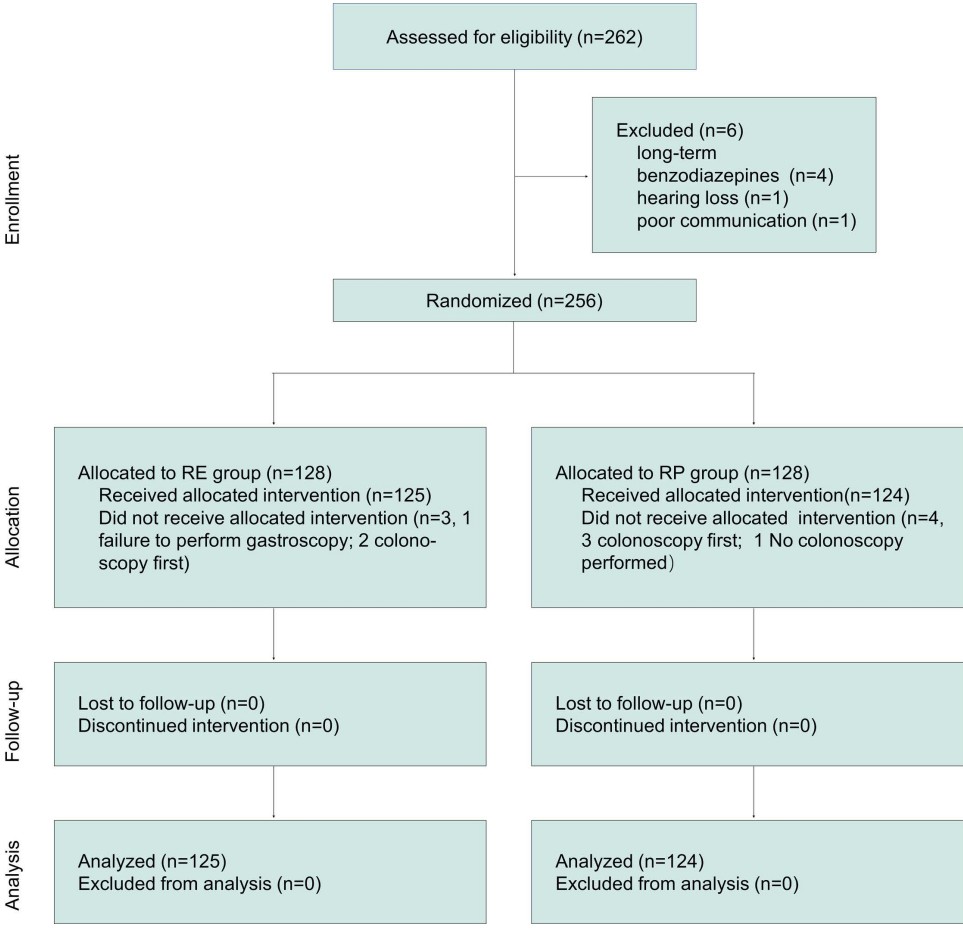

**Fig 1.  CONSORT flowchart of patient enrollment, randomization, and follow-up.** RE: remimazolam-etomidate, RP: remimazolam-propofol.

**Table 1. Baseline patient characteristics.**

| Variable | RE group (n = 125) | RP group (n = 124) | t/z/χ² | P |
|---|---|---|---|---|
| Age, M(P25,P75) | 58(47,65) | 55.5(43,61.75) | z = 1.54 | 0.125 |
| Sex | | | χ² = 0.68 | 0.410 |
| Male, n(%) | 59(47.2) | 65(52.4) | | |
| Female, n(%) | 66(52.8) | 59(47.6) | | |
| Height, Mean±SD | 167.3±8.1 | 168.4±7.8 | t = 1.13 | 0.439 |
| Weight, Mean±SD | 69.2±11.3 | 70.4±12.1 | t = 0.79 | 0.650 |
| BMI, Mean±SD | 24.6±3.0 | 24.7±3.1 | t = 0.17 | 0.920 |
| ASA | | | χ² = 1.87 | 0.393 |
| I, n(%) | 18(14.4) | 26(21.0) | | |
| II, n(%) | 104(83.2) | 95(76.6) | | |
| III, n(%) | 3(2.4) | 3(2.4) | | |
| Mallampati | | | χ² = 0.09 | 0.955 |
| I, n(%) | 6(4.8) | 7(5.6) | | |
| II, n(%) | 109(87.2) | 107(86.3) | | |
| III, n(%) | 10(8.0) | 10(8.1) | | |
| Smoking, n(%) | 34(27.2) | 38(30.6) | χ² = 0.36 | 0.549 |
| Alcohol, n(%) | 35(28.0) | 34(27.4) | χ² = 0.10 | 0.918 |
| Operation history, n(%) | 27(21.6) | 20(16.1) | χ² = 1.22 | 0.270 |
| Comorbidities, n(%) | 43(34.4) | 36(29.0) | χ² = 0.83 | 0.363 |
| Sleep apnea history, n(%) | 23(18.4) | 26(21.0) | χ² = 0.26 | 0.610 |
| PRODIGY score, M(P25,P75) | 11(8,16) | 11(4.25,16) | z = 0.73 | 0.468 |
| Low risk, n(%) | 23(18.4) | 31(25.0) | z = 1.92 | 0.382 |
| Intermediate risk, n(%) | 59(47.2) | 55(40.3) | | |
| High risk, n(%) | 43(34.4) | 38(34.7) | | |
| Procedure time,min, M(P25, P75) | 32(27,38) | 31(26,35) | z = 1.49 | 0.992 |
| Sedation time, min, M(P25, P75) | 40(35,48) | 40(35,44.75) | z = 1.22 | 0.221 |
| Awake time,min, M(P25, P75) | 4(4,5) | 4(4,4) | z = 1.26 | 0.208 |
| Administered dose of Sedatives | | | | |
| Remimazolam,mg, Mean±SD | 23.4±4.5 | 23.2±4.1 | t = 0.46 | 0.514 |
| Etomidate,mg, M(P25, P75) | 7(6,7.5) | – | | |
| Propofol,mg, M(P25, P75) | – | 50(45,60) | | |
| Sufentanil,mcg, M(P25, P75) | 5(5,5) | 5(5,5) | z = 0.86 | 0.382 |
| Flumazenil,mg, M(P25, P75) | 0.3(0.3,0.3) | 0.3(0.3,0.3) | z = 1.62 | 0.106 |
| Volume of fluid,mL, M(P25, P75) | 500(500,500) | 500(500,500) | z = 0.25 | 0.807 |

BMI: Body Mass Index, ASA: American Society of Anesthesiologists, SD: Standard Deviation.

the two groups. Etomidate was used at 7(6,7.5) mg in the RE group and propofol was used at 50(45,60) mg in the RP group. Fig 2 depicts the variations in vital signs recorded in relation to elapsed sedation time.

## Primary and secondary outcomes

The incidence of respiratory depression was significantly lower in the RE group (20.0%, 25/125) compared to the RP group (32.3%, 40/124; OR=0.52, 95% CI = 0.29 to 0.93, *p* = 0.028) (Table 2). Respiratory depression subgroup analyses were carried out in predefined subgroups (Fig 3). The incidence of respiratory depression appeared to be lower in the RE group than in the RP group in the following populations: males (OR,0.38; 95% CI, 0.17 to 0.84, *p* = 0.019),

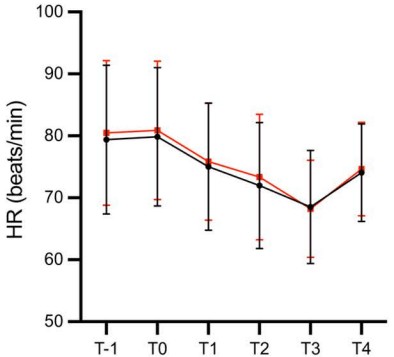
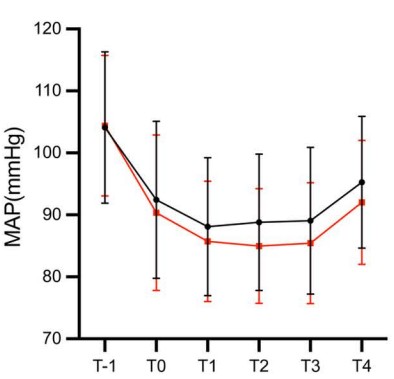
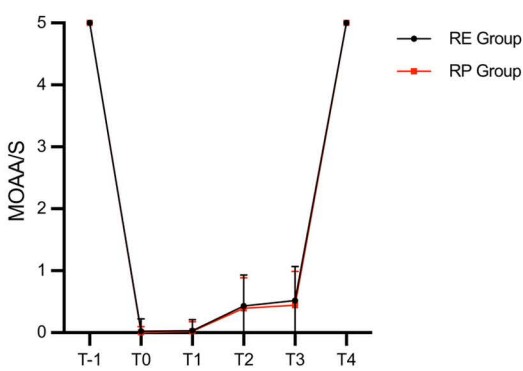

**Fig 2. Changes in vital signs against elapsed sedation time.** MAP, Mean arterial pressure; HR, Heart rate; MOAA/S: Modified observer's assessment of alertness/sedation scale. T-1: On entry to the examination room; T0: When placing the gastroscope; T1: three minutes after placement of the gastroscope; T2: When placing the colonoscope; T3: three minutes after colonoscopy placement; T4: On exit from the examination room.

**Table 2. Primary and secondary outcomes.**

| Variable | RE group (n = 125) | RP group (n = 124) | t/z/χ² | P |
|---|---|---|---|---|
| Respiratory depression, n(%) | 25(20.0) | 40(32.3) | χ²=4.85 | 0.028 |
| Hypoxemia, n(%) | 3(2.4) | 5(4.0) | χ²=0.14 | 0.711 |
| Airway intervention | 94(75.2) | 105(84.7) | χ²=3.48 | 0.062 |
| The numbers of airway intervention | | | χ²=8.13 | 0.043 |
| 0, n(%) | 31(24.8) | 19(15.3) | | |
| 1, n(%) | 60(48.0) | 56(45.2) | | |
| 2, n(%) | 27(21.6) | 31(25.0) | | |
| 3, n(%) | 7(5.6) | 18(14.5) | | |
| Bag-Mask Ventilation, n(%) | 3(2.4) | 5(4.0) | χ²=0.12 | 0.728 |
| Removal of endoscope due to Hypoxia, n(%) | 3(2.4) | 5(4.0) | χ²=0.12 | 0.728 |
| Successful sedation, n(%) | 123(98.4) | 124(100.0) | Fisher | 0.498 |
| Number of remedies, n(%) | 2(1.6) | 0(0) | Fisher | 0.498 |

age > 59 years(OR,0.35; 95% CI, 0.13 to 0.87, $p = 0.026$), BMI < 25 kg/m²(OR,0.17; 95% CI, 0.04 to 0.57, $p = 0.008$), with comorbidities(OR,0.23; 95% CI, 0.08 to 0.59, $p = 0.003$), sleep apnea(OR,0.13; 95% CI, 0.02 to 0.59, $p = 0.016$), PRODIGY as high risk(OR,0.28; 95% CI, 0.11 to 0.69, $p = 0.006$). Multivariable logistic regression (S1 Table), adjusting for age, sex, BMI, ASA classification, comorbidities, sleep apnea history, and PRODIGY score, confirmed that the RE group had a lower incidence of respiratory depression compared to the RP group (OR=0.35, 95% CI: 0.16 to 0.77, $p = 0.009$).

About secondary outcomes (Table 2), hypoxemia occurred in three patients (2.4%) in the RE group and five patients (4.0%) in the RP group. Similarly, endoscope removal due to hypoxemia was required in three patients (2.4%) in the RE group and five patients (4.0%) in the RP group. No significant differences were observed between the two groups in the incidence of hypoxemia ($p = 0.728$) or the frequency of endoscope removal due to hypoxemia ($p = 0.728$). Airway interventions were performed in 94 patients (75.2%) in the RE group and 105 patients (84.7%) in the RP group, with no statistically significant difference ($p = 0.062$). However, a statistically significant difference was noted in the distribution of the number of airway interventions between the groups ($p = 0.043$), with 18 patients (14.5%) in the RP group requiring three interventions compared to only seven patients (5.6%) in the RE group. Remedial sedation was administered to two

| Subgroup | RP Group | RE Group | | OR (95% CI) | P value | P for interaction |
|---|---|---|---|---|---|---|
| Overall | 40/124 (32.3) | 25/125 (20.0) | | 0.52 (0.29–0.93) | 0.029 | |
| Sex | | | | | | 0.228 |
| Male | 26/65 (40.0) | 12/59 (20.3) | | 0.38 (0.17–0.84) | 0.019 | |
| Female | 14/59 (23.7) | 13/66 (19.7) | | 0.79 (0.33–1.86) | 0.585 | |
| Age | | | | | | 0.278 |
| 18–59 | 24/83 (28.9) | 15/70 (21.4) | | 0.67 (0.31–1.40) | 0.291 | |
| >59 | 16/41 (39.0) | 10/55 (18.2) | | 0.35 (0.13–0.87) | 0.026 | |
| ASA | | | | | | 0.665 |
| I–II | 38/121 (31.4) | 24/122 (19.7) | | 0.53 (0.29–0.96) | 0.037 | |
| III | 2/3 (66.7) | 1/3 (33.3) | | 0.25 (0.00–6.56) | 0.423 | |
| BMI | | | | | | 0.085 |
| <25 | 14/65 (21.5) | 3/66 (4.5) | | 0.17 (0.04–0.57) | 0.008 | |
| 25–30 | 19/50 (38.0) | 19/55 (34.5) | | 0.86 (0.39–1.91) | 0.713 | |
| >30 | 7/9 (77.8) | 3/4 (75.0) | | 0.86 (0.06–22.75) | 0.913 | |
| Comorbidities | | | | | | 0.057 |
| No | 19/89 (21.3) | 14/82 (17.1) | | 0.76 (0.35–1.63) | 0.48 | |
| Yes | 21/35 (60.0) | 11/43 (25.6) | | 0.23 (0.08–0.59) | 0.003 | |
| Sleep apnea | | | | | | 0.099 |
| No | 16/98 (16.3) | 11/102 (10.8) | | 0.62 (0.27–1.40) | 0.255 | |
| Yes | 24/26 (92.3) | 14/23 (60.9) | | 0.13 (0.02–0.59) | 0.016 | |
| PRODIGY | | | | | | 0.463 |
| Low risk(<8) | 3/31 (9.7) | 1/23 (4.3) | | 0.42 (0.02–3.57) | 0.471 | |
| Intermediate risk(8–14) | 12/55 (21.8) | 9/59 (15.3) | | 0.64 (0.24–1.67) | 0.368 | |
| High risk(>14) | 25/38 (65.8) | 15/43 (34.9) | | 0.28 (0.11–0.69) | 0.006 | |

*no. of events / total no. (%)*

0.1 1 7.4

RE Better RP Better

**Fig 3. Subgroup analyses of respiratory depression.** BMI: Body Mass Index, ASA: American Society of Anesthesiologists, OR, Odds ratio, RE: remimazolam-etomidate, RP: remimazolam-propofol.

patients in the RE group, yielding a sedation success rate of 98.4%, while no patients in the RP group required remedial sedation, achieving a success rate of 100% ($p = 0.498$).

### Adverse events and postoperative follow up

Table 3 summarizes the adverse events observed during gastrointestinal endoscopy. Hypotension occurred in 23.2% of patients sedated with RE compared to 36.3% of those sedated with RP, a difference that was statistically significant ($p = 0.024$). Among those experiencing hypotension, seven patients (5.6%) in the RE group and 12 patients (9.7%) in the RP group required intervention, though this difference was not statistically significant ($p = 0.226$). Bradycardia was observed in five patients (4.0%) in both the RE and RP groups, with no significant difference ($p = 0.990$). Tachycardia was noted in three patients (2.4%) sedated with RE, but in none of those sedated with RP ($p = 0.247$). No instances of injection pain were reported; however, muscle tremor occurred in two patients (1.6%) in the RE group ($p = 0.498$).

In the post-anesthesia care unit (PACU), nausea and vomiting were reported in 13 patients (10.4%) in the RE group and 14 patients (11.3%) in the RP group, with no significant difference ($p = 0.821$). On the first postoperative day, nausea and vomiting occurred in one patient (0.8%) in the RE group and three patients (2.4%) in the RP group, also showing no significant difference ($p = 0.622$). Both endoscopists ($p = 0.216$) and patients ($p = 0.138$) expressed satisfaction with the gastrointestinal endoscopy procedure.

### Discussion

This randomized clinical trial assessed the efficacy and safety of combining various sedative agents in patients undergoing gastrointestinal endoscopic procedures. The findings demonstrated that the incidence of respiratory depression and

**Table 3. Adverse events and postoperative follow up.**

| Variable | RE group (n = 125) | RP group (n = 124) | t/z/χ² | P |
|---|---|---|---|---|
| Nausea and vomiting | | | | |
| Postoperative, n(%) | 13(10.4) | 14(11.3) | $\chi^2 = 0.05$ | 0.821 |
| Postoperative Day 1, n(%) | 1(0.8) | 3(2.4) | Fisher | 0.622 |
| Hypotension, n(%) | 29(23.2) | 45(36.3) | $\chi^2 = 5.10$ | 0.024 |
| Hypotension treated, n(%) | 7(5.6) | 12(9.7) | $\chi^2 = 1.47$ | 0.226 |
| Bradycardia, n(%) | 5(4.0) | 5(4.0) | $\chi^2 = 0.00$ | 0.990 |
| Tachycardia | 3(2.4%) | 0(0) | Fisher | 0.247 |
| Satisfaction for sedation | | | | |
| Endoscopist, M(P25, P75) | 9(8,9) | 9(8,9) | z = 1.24 | 0.216 |
| Patient, M(P25, P75) | 9(8,9) | 9(8,9) | z = 1.49 | 0.138 |

RE: remimazolam-etomidate, RP: remimazolam-propofol

hypotension was lower with the combination of remimazolam and etomidate compared to remimazolam and propofol during sedation.

Procedural sedation during gastrointestinal endoscopy enhances patient comfort and alleviates anxiety [2]. Historically, propofol has been the most widely used and effective sedative agent [11]. However, its potent effects are marred by respiratory and circulatory depression, as well as injection pain [4]. Studies have shown that a higher proportion of patients experience respiratory depression [12,13], hypotension [14,15], and injection pain [16] when using propofol for procedural sedation. Remimazolam, a benzodiazepine, binds to central gamma-aminobutyric acid A receptors inducing a sedative-amnesic effect [17]. Studies have shown that compared with propofol, remimazolam did not cause injection pain and reduced cardiovascular and respiratory inhibition [18,19]. The sedative effect of remimazolam can quickly be reversed by flumazenil, and it has higher safety in clinical applications [20]. Remimazolam has been regarded as a potential better prospect in gastrointestinal endoscopy sedation [21]. However, failure of first sedation exists with a single fixed dose of remimazolam administered according to the drug's instructions. Therefore, most current studies use weight-based administration for sedation induction. The study [22] demonstrated that the sedation success rate with 0.2 mg/kg remimazolam was non-inferior to that with 1.5 mg/kg propofol, however, 0.15 mg/kg remimazolam failed to meet the non-inferiority threshold. Due to the suboptimal success rate of initial induction with remimazolam alone [22], combining propofol and etomidate may be a viable alternative. Another study [16] reported that the combination of remimazolam and propofol resulted in fewer adverse events compared to propofol monotherapy, while also offering improved sedative efficacy and greater endoscopist satisfaction than remimazolam monotherapy. And, study [23] found that the remimazolam monotherapy regimen did not demonstrate superior safety or efficacy compared to the etomidate-propofol combination for gastrointestinal endoscopy. Building upon these findings, our study provides insights into the performance of remimazolam combined with etomidate relative to its combination with propofol.

In our study, respiratory depression occurred in 20% of patients in the RE group, reflecting a 38% reduction in such events and a decreased need for airway maneuvers compared to the RP group. Furthermore, the incidence of hypotension was lower in the RE group, underscoring its superior safety profile and reduced sedation-related adverse effects. Our sensitivity analyses reinforce the primary results, demonstrating that the reduced respiratory depression with remimazolam-etomidate persists after adjusting for potential confounders. However, an exploratory subgroup analysis of respiratory depression incidence revealed that not all outcomes favored the RE group, urging clinicians to carefully evaluate factors such as patient sex, age, BMI, comorbidities, and sleep apnea when determining the most suitable sedative agents. Our findings lend stronger support to the use of remimazolam combined with etomidate for sedation in older patients with comorbidities and an elevated risk of respiratory depression. Notably, injection pain was absent in both groups, possibly

due to the administration of propofol following remimazolam, when patients were already sedated. However, the occurrence of muscle tremors in two patients, despite the sequential use of remimazolam and etomidate, suggests that remimazolam may be less effective than propofol in suppressing etomidate-induced muscle tremors.

The elevated incidence of respiratory depression observed in our study, compared to prior studies [16,24], may be attributed to the higher doses of sedative medications administered in our protocol. In our clinical practice, adhering to the recommended doses from previous studies often resulted in endoscopists being unable to insert the scope, thereby hindering subsequent procedures. Consequently, we opted for higher doses to achieve effective sedation. Accordingly, the duration of deep sedation (see Fig 2) in our study exceeded that reported in earlier studies [16]. The incidence of hypoxemia in our study aligned with findings from previous research [13,25], hovering around 10%. Patient awakening times were consistent with those reported in prior studies [7,26] and showed no prolongation. This can be attributed to the use of flumazenil, which effectively reverses the residual effects of remimazolam. The availability of a specific antagonist for remimazolam, unlike propofol, enhances its safety profile and supports its use in procedural sedation. Additionally, although both propofol and remimazolam exhibit rapid onset and offset of action, the short offset of propofol stems from its redistribution into lipid compartments rather than swift metabolism [27]. Overall, the combination of remimazolam and etomidate proved safer, with a lower incidence of adverse events compared to the combination with propofol. These findings offer a practical and viable option for selecting sedative agents in procedural sedation.

Our findings should be interpreted in light of several limitations. First, our sample size calculation assumed a 50% relative reduction in respiratory depression, which may overestimate the expected effect size. While our results demonstrated a statistically significant difference, the generalizability of these findings would benefit from validation in larger, multicenter studies powered for smaller effect sizes. Second, our study did not systematically assess post-procedural pain scores in the PACU or upon discharge, which could have provided valuable insights into patient recovery and comfort. Third, this study utilized sufentanil as the analgesic, which is known to elevate the risk of respiratory depression; future studies could explore the novel analgesic oliceridine [28] to potentially lower this incidence.

In conclusion, this randomized controlled trial compared the use of etomidate and propofol in combination with remimazolam for patients undergoing gastrointestinal endoscopy under procedural sedation. The results revealed a significantly lower incidence of respiratory depression with remimazolam-etomidate compared to remimazolam-propofol. Additionally, remimazolam-etomidate demonstrated certain hemodynamic benefits over remimazolam-propofol. Nonetheless, determining the optimal dosage requires additional investigation, and exploring combinations with novel analgesic agents offers a compelling direction for future studies.

## Supporting information

**S1 Table. Multivariable logistic regression for respiratory depression.**
(DOCX)

**S1 Protocol. This is the English version of the research proposal.**
(PDF)

**S2 Protocol. This is the Chinese version of the research proposal.**
(PDF)

## Acknowledgments

We acknowledge all the staff members for their contributions to this study. Special thanks are extended to all the patients who participated in this research.

## Author contributions

**Conceptualization:** Baoyu Ma, Shoushi Wang.

**Data curation:** Shoushi Wang.

**Formal analysis:** Baoyu Ma.

**Investigation:** Ning Zhang, Rong Huang.

**Methodology:** Baoyu Ma, Shoushi Wang.

**Project administration:** Shoushi Wang.

**Resources:** Shoushi Wang.

**Software:** Baoyu Ma.

**Supervision:** Shoushi Wang.

**Validation:** Shoushi Wang.

**Visualization:** Baoyu Ma.

**Writing – original draft:** Baoyu Ma, Ning Zhang.

**Writing – review & editing:** Shoushi Wang.

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
