## [Decision Letter · Decision Letter 0]

PONE-D-25-09878
Remimazolam-Etomidate versus Remimazolam-Propofol for Gastrointestinal Endoscopy: A Randomized Controlled Trial
PLOS ONE

Dear Dr. Wang,

Thank you for submitting your manuscript to PLOS ONE. After careful consideration, we feel that it has merit but does not fully meet PLOS ONE’s publication criteria as it currently stands. Therefore, we invite you to submit a revised version of the manuscript that addresses the points raised during the review process.

We look forward to receiving your revised manuscript.

Kind regards,

Chih-Wei Tseng

Academic Editor

PLOS ONE

Journal Requirements:

2. We note that you have selected “Clinical Trial” as your article type. PLOS ONE requires that all clinical trials are registered in an appropriate registry (the WHO list of approved registries is at https://www.who.int/clinical-trials-registry-platform/network/primary-registries" https://www.who.int/clinical-trials-registry-platform/network/primary-registries and more information on trial registration is at http://www.icmje.org/about-icmje/faqs/clinical-trials-registration/). Please state the name of the registry and the registration number (e.g. ISRCTN or ClinicalTrials.gov) in the submission data and on the title page of your manuscript. a) Please provide the complete date range for participant recruitment and follow-up in the methods section of your manuscript. b) If you have not yet registered your trial in an appropriate registry, we now require you to do so and will need confirmation of the trial registry number before we can pass your paper to the next stage of review. Please include in the Methods section of your paper your reasons for not registering this study before enrolment of participants started. Please confirm that all related trials are registered by stating: “The authors confirm that all ongoing and related trials for this drug/intervention are registered”. Please see http://journals.plos.org/plosone/s/submission-guidelines#loc-clinical-trials for our policies on clinical trials.

4. 
Please remove all personal information, ensure that the data shared are in accordance with participant consent, and re-upload a fully anonymized data set.

Reviewers' comments:

Reviewer's Responses to Questions

**Comments to the Author**

1. Is the manuscript technically sound, and do the data support the conclusions?

Reviewer #1: Yes

Reviewer #2: Yes

Reviewer #3: Yes

2. Has the statistical analysis been performed appropriately and rigorously? 

Reviewer #1: Yes

Reviewer #2: Yes

Reviewer #3: No

3. Have the authors made all data underlying the findings in their manuscript fully available?

Reviewer #1: Yes

Reviewer #2: Yes

Reviewer #3: No

4. Is the manuscript presented in an intelligible fashion and written in standard English?

Reviewer #1: Yes

Reviewer #2: Yes

Reviewer #3: Yes

5. Review Comments to the Author

Reviewer #1: The authors present an interesting topic looking at the combination of Remimazolam-Etomidate versus Remimazolam-Propofol. The authors found that the RP group had significantly higher incidence of hypotension and respiratory depression vs. the RE group.

The authors do not look into the post-procedure pain of patients which would be very helpful to include - perhaps looking into adding pain scores of the patients in PACU or upon discharge.

In the USA, versed is still the gold standard sedative medication. Including versed-etomidate/versed-propofol groups for comparison would be very helpful as well.

Who was monitoring the administration of the anesthesia? Was this CRNAs or board-certified anesthesiologists? This information should be included in the manuscript.

Reviewer #2: As the statistical reviewer I will focus on methods and reporting.

Major

1) the power of the study is limited - the RCT is powered at 80% to observe a 50% reduction in the primary outcome. this needs to be discussed as a major limitation.

2) Broadly there is balance in the two groups, as evidenced by table 1 - however there are some key differences for example age. can the authors conduct sensitivity analyses where they control for age and other characteristics where there is imbalance in a regression framework (logistic for binary, linear for continuous outcomes).g

Minor

1) please expand the methods section of the abstract to briefly state the analytical methods used.

2) Explain the strategy for dealing with misisng data at the end of the statistical analyses section. were all data complete for all patients?

Reviewer #3: 1- what is the justification for selecting this number of community reporters?

2- bias might be there if you have recruited members form local company especially if they know about it before?

3- you cant compare six months recall period by community informants vs 12 months from WHO

4- clarify hoe duplicated reports are handled

5- don't used under estimation in page 12 without any strong validation and bench marking

6- check reptation and grammar mistakes and terminology as well

7- add more recent literature review

8- some figure caption lacks sufficient details

9-what about safety and confidentiality? explain

10- add statistics measures

6. PLOS authors have the option to publish the peer review history of their article (what does this mean?). If published, this will include your full peer review and any attached files.

Reviewer #1: No

Reviewer #2: No

Reviewer #3: **Yes: **Dr. Yousra Nomeir

---

## [Author Response · Author response to Decision Letter 1]

9 May 2025

We are grateful for the opportunity to revise our work and address the comments provided by the editor and reviewers. We have carefully considered all feedback and made the necessary revisions to enhance the quality and clarity of our manuscript. Below, we respond to each comment individually and detail the corresponding changes.

Reviewer #1:

Q:The authors present an interesting topic looking at the combination of Remimazolam-Etomidate versus Remimazolam-Propofol. The authors found that the RP group had significantly higher incidence of hypotension and respiratory depression vs. the RE group.

A:Thank you for your time and valuable comments on our manuscript. We appreciate your positive feedback on our study comparing remimazolam-etomidate (RE) and remimazolam-propofol (RP) for sedation during gastrointestinal endoscopy. We agree with your observation. Our results showed that RE was associated with a lower incidence of respiratory depression and hypotension compared to RP. These findings support RE as a safer alternative for procedural sedation, particularly in high-risk patients.

Q:1.The authors do not look into the post-procedure pain of patients which would be very helpful to include - perhaps looking into adding pain scores of the patients in PACU or upon discharge.

A:We sincerely appreciate your thoughtful feedback and valuable suggestion regarding the assessment of post-procedure pain in our study. We fully agree that evaluating pain scores in the post-anesthesia care unit (PACU) or upon discharge would have provided additional insights into patient comfort and the overall efficacy of the sedation regimens.

However, as you rightly pointed out, this metric was not included in our original study design or data collection protocol. While we recognize the clinical relevance of post-procedural pain assessment, the absence of these data prevents us from incorporating such an analysis in the current manuscript. To address this limitation transparently, we have added a statement in the Limitations section (Page 19, Lines 13–14) acknowledging that future studies should include standardized pain scores to further evaluate patient-centered outcomes.

Thank you again for your constructive comment, which highlights an important area for improvement in subsequent research. We greatly value your expertise and hope this clarification aligns with your expectations.

Q:2.In the USA, versed is still the gold standard sedative medication. Including versed-etomidate/versed-propofol groups for comparison would be very helpful as well.

A:We sincerely appreciate your insightful suggestion regarding the inclusion of midazolam-based sedation groups (midazolam-etomidate or midazolam-propofol) for comparative analysis. We fully acknowledge the clinical relevance of midazolam as a gold standard sedative in the USA and other regions.

However, our study design was guided by current clinical practice trends in China and several East Asian countries (e.g., Japan, South Korea), where remimazolam has been approved for gastrointestinal endoscopy and is increasingly replacing midazolam. This shift is supported by Phase II/III clinical trials demonstrating remimazolam’s superior safety profile, rapid recovery, compared to midazolam [1-3]. Notably, remimazolam’s pharmacokinetic advantages (e.g., short half-life, flumazenil reversibility) align with the demand for efficient procedural sedation in high-volume endoscopy centers.

While we agree that direct comparisons with midazolam-based regimens would be valuable, our institutional protocols and regional guidelines prioritize remimazolam-based combinations. We recognize this as a potential limitation for generalizability to settings where midazolam remains predominant. Future multicenter studies incorporating midazolam comparators could further bridge this gap.

Q:3.Who was monitoring the administration of the anesthesia? Was this CRNAs or board-certified anesthesiologists? This information should be included in the manuscript.

A:Thank you for your valuable question regarding anesthesia administration in our study. We appreciate the opportunity to clarify this important methodological detail.

In response to your query, all procedural sedation in this trial was administered and monitored exclusively by board-certified anesthesiologists. Specifically, continuous physiological monitoring was performed by attending anesthesiologists who had completed standardized training in endoscopic sedation protocols. We recognize that specifying the qualifications of anesthesia providers is crucial for study reproducibility and clinical interpretation. Accordingly, we have added this clarification to the Methods section (Page 6, Lines 19-22).

Thank you for highlighting this important aspect of our study methodology.

Reviewer #2:

As the statistical reviewer I will focus on methods and reporting.

Major

Q:1) the power of the study is limited - the RCT is powered at 80% to observe a 50% reduction in the primary outcome. this needs to be discussed as a major limitation.

A:We sincerely appreciate the reviewer’s insightful comment regarding the statistical power of our study.

We acknowledge the reviewer’s valid point about the study’s power and the magnitude of the effect size assumed in our sample size calculation. As noted, our study was designed with 80% power to detect a 50% relative reduction in the incidence of respiratory depression (from 30% in the remimazolam-propofol [RP] group to 15% in the remimazolam-etomidate [RE] group). While this assumption was based on pre-trial data and previous literature, we agree that the chosen effect size may be considered optimistic, and we have now explicitly addressed this as a limitation in the revised manuscript (Page19, Lines 7-11): our sample size calculation assumed a 50% relative reduction in respiratory depression, which may overestimate the expected effect size. While our results demonstrated a statistically significant difference, the generalizability of these findings would benefit from validation in larger, multicenter studies powered for smaller effect sizes.

We hope this clarification addresses the reviewer’s concern. Thank you for highlighting this important aspect, which strengthens the transparency of our work.

Q:2) Broadly there is balance in the two groups, as evidenced by table 1 - however there are some key differences for example age. can the authors conduct sensitivity analyses where they control for age and other characteristics where there is imbalance in a regression framework (logistic for binary, linear for continuous outcomes).g

A:We sincerely appreciate the reviewer's insightful suggestion regarding the potential impact of baseline imbalances on our study outcomes. In response to this valuable comment, we have conducted additional sensitivity analyses to rigorously evaluate the robustness of our primary findings.

We performed multivariable logistic regression analyses for the primary outcome (respiratory depression), adjusting for the following covariates where clinically relevant: age, sex, BMI, ASA, comorbidities, sleep apnea history and PRODIGY score. The adjusted model demonstrated that the remimazolam-etomidate (RE) group maintained a significantly lower risk of respiratory depression compared to the remimazolam-propofol (RP) group (adjusted OR=0.35, 95% CI: 0.16-0.77, p=0.009), consistent with our primary unadjusted analysis (OR=0.52, 95% CI: 0.29-0.93, p=0.028).

These sensitivity analyses confirm that our primary findings remain robust after accounting for potential confounding from baseline characteristics. The consistent treatment effect across both unadjusted and adjusted models strengthens the validity of our conclusion that remimazolam-etomidate combination demonstrates a superior respiratory safety profile compared to remimazolam-propofol for gastrointestinal endoscopy sedation.

We have added these results to the revised manuscript in the Results section (Page 13，Lines 4-7) and included the full model details in the Methods section (Page 9，Lines 7-10) and Supplementary Materials. The Discussion section (Page 17，Lines 20-22) has been updated to reflect these additional analyses. We believe these additional analyses have strengthened our manuscript and thank the reviewer for this constructive suggestion.

Minor

Q:1) please expand the methods section of the abstract to briefly state the analytical methods used.

A:We sincerely appreciate the reviewer’s constructive feedback. In accordance with the journal’s abstract word limit (300 words), we have expanded the methods section (Page 2，Lines 11-13) of the abstract to concisely include key statistical methods.

Specifically, we added the following sentence: "Statistical analyses included t-tests, Mann-Whitney U tests, and χ² tests for group comparisons, with subgroup analyses and multivariable logistic regression to assess the robustness of primary outcome." This addition highlights our analytical approach while adhering to the journal’s word constraints. The subgroup and sensitivity analyses underscore the clinical relevance and consistency of our findings.

Thank you for your valuable suggestion, which has strengthened our manuscript.

Q:2) Explain the strategy for dealing with missing data at the end of the statistical analyses section. were all data complete for all patients?

A:We sincerely appreciate the reviewer’s insightful comment regarding the handling of missing data in our study.

All data collected for this study, particularly those pertaining to the primary and secondary outcomes (e.g., incidence of respiratory depression, hypoxemia, hemodynamic stability, and procedural success rates), were complete for all enrolled patients. No missing data were encountered for these critical endpoints, ensuring the robustness of our statistical analyses and conclusions.

We hope this clarification addresses the reviewer’s concern.

Reviewer #3:

We sincerely appreciate your thorough review and valuable feedback on our manuscript. While some of the points raised appear to relate to aspects not directly addressed in our study (e.g., recall periods), we recognize the importance of methodological clarity and have carefully reviewed all comments to ensure our responses align with the study’s scope. Below are my detailed responses to the comments. We hope these revisions meet your expectations. Should any concerns remain, we are fully committed to further refining the manuscript. Thank you for your time and expertise in improving our work.

Q:1- what is the justification for selecting this number of community reporters?

A:We appreciate the reviewer’s insightful question regarding the sample size calculation in our study.

The sample size was calculated based on the primary outcome (respiratory depression), assuming a 30% incidence in the remimazolam-propofol (RP) group and a 50% relative reduction with remimazolam-etomidate (RE). With 80% power and a 5% significance level, 118 patients per group were required. Accounting for a 5% dropout rate, we enrolled 124 patients per group (total 248). This aligns with similar sedation studies and ensures robust detection of clinically meaningful differences.

We hope this clarification addresses the reviewer’s concern.

Q:2- bias might be there if you have recruited members form local company especially if they know about it before?

A:We appreciate the reviewer’s concern regarding potential bias in participant recruitment.

We would like to clarify that all participants in this study were patients routinely scheduled for gastrointestinal endoscopy at our hospital, with no prior knowledge of their group assignment. The randomization process was computer-generated and strictly implemented to ensure unbiased allocation to either the remimazolam-etomidate or remimazolam-propofol groups. Importantly, the study employed a double-blind design where both participants and clinicians were unaware of group assignments throughout the trial period.

We believe these measures effectively address the reviewer's concern regarding recruitment and allocation bias in our trial.

Q:3- you can’t compare six months recall period by community informants vs 12 months from WHO

A:Thank you for your valuable feedback.

Our study is a prospective randomized controlled trial, with all data (e.g., respiratory depression, hypotension) collected in real-time during the procedure and immediate post-op period (≤24h). No retrospective or long-term follow-up data were included.

We are happy to revise the manuscript to address any specific concerns. Please let us know if further clarification is needed.

Q:4- clarify hoe duplicated reports are handled

A:Thank you for your valuable feedback regarding the handling of duplicated reports in our study. We appreciate your attention to methodological rigor. To clarify our approach:

Prevention during enrollment: each participant was assigned a unique study ID.

Data collection and management: case report forms included signature fields by anesthesiologists to verify procedural uniqueness.

Statistical analysis: no duplicates were detected in the final dataset.

If the above response fails to address your query, it may be due to our incomplete understanding of your question. We kindly encourage you to provide more detailed information, as we are committed to refining the academic quality of our content based on your feedback.

Q:5- don't used under estimation in page 12 without any strong validation and bench marking

A:Thank you for your valuable feedback on our manuscript. We sincerely appreciate the time and effort you have dedicated to reviewing our work. Regarding your comment about "under estimation in page 12 without any strong validation and benchmarking," we acknowledge that this point was not entirely clear to us, as our manuscript does not explicitly mention "under estimation" on page 12 or elsewhere.

If we have misinterpreted your comment, we would be grateful for further clarification regarding the specific aspect of "under estimation" you are referring to. This will allow us to provide a more targeted response or make the necessary revisions to the manuscript.

Q:6- check repetition and grammar mistakes and terminology as well

A:Thank you for your valuable feedback. We have carefully revised the manuscript to address repetition, grammar, and terminology issues. Key changes include:

Repetition: Removed redundant descriptions in the Methods and Results sections.

Grammar: Corrected errors in verb agreement, articles, and sentence structure.

Terminology: Standardized terms (e.g., "SpO2" instead of "SPO2").

The revised manuscript is now clearer and more concise. We appreciate your time and hope these changes meet your expectations.

Q:7- add more recent literature review

A:Thank you for your valuable feedback regarding the need to incorporate more recent literature into our manuscript. We fully agree that updating the references would enhance the paper's relevance and scholarly impact. In response to your suggestion, we have carefully reviewed and added several key recent publications (Reference15, 23) to strengthen the literature review and contextualize our findings.

These additions ensure our manuscript reflects the latest advancements in procedural sedation while maintaining focus on the original study objectives.

Q:8- some figure caption lacks sufficient details

A:Thank you for your valuable feedback regarding the insufficient details in some figure captions. We have carefully revised the captions for all figures, particularly Figure 1, to provide clearer and more comprehensive descriptions.

Thank you again for your constructive critique.

Q:9- what about safety and confidentiality? Explain

A:Thank you for your valuable feedback regarding our manuscript. We appreciate your time and effort in reviewing our work. Regarding your comment about "safety and confidentiality," we would like to clarify that our study primarily focused on comparing the efficacy and safety profiles of remimazolam-etomidate versus remimazolam-propofol for procedural sedation during gastrointestinal endoscopy.

In our manuscript, we have thoroughly addressed safety by reporting detailed adverse events, including respiratory depression, hypotension, bradycardia,

---

## [Decision Letter · Decision Letter 1]

Remimazolam-Etomidate versus Remimazolam-Propofol for Gastrointestinal Endoscopy: A Randomized Controlled Trial

PONE-D-25-09878R1

Dear Dr. Wang,

We’re pleased to inform you that your manuscript has been judged scientifically suitable for publication and will be formally accepted for publication once it meets all outstanding technical requirements.

Kind regards,

Chih-Wei Tseng

Academic Editor

PLOS ONE

Additional Editor Comments (optional):

Reviewers' comments:

Reviewer's Responses to Questions

**Comments to the Author**

1. If the authors have adequately addressed your comments raised in a previous round of review and you feel that this manuscript is now acceptable for publication, you may indicate that here to bypass the “Comments to the Author” section, enter your conflict of interest statement in the “Confidential to Editor” section, and submit your "Accept" recommendation.

Reviewer #1: All comments have been addressed

Reviewer #2: All comments have been addressed

Reviewer #3: All comments have been addressed

2. Is the manuscript technically sound, and do the data support the conclusions?

Reviewer #1: (No Response)

Reviewer #2: Yes

Reviewer #3: Yes

3. Has the statistical analysis been performed appropriately and rigorously? 

Reviewer #1: (No Response)

Reviewer #2: (No Response)

Reviewer #3: Yes

4. Have the authors made all data underlying the findings in their manuscript fully available?

Reviewer #1: (No Response)

Reviewer #2: Yes

Reviewer #3: Yes

5. Is the manuscript presented in an intelligible fashion and written in standard English?

Reviewer #1: (No Response)

Reviewer #2: Yes

Reviewer #3: Yes

6. Review Comments to the Author

Reviewer #1: (No Response)

Reviewer #2: I am satisfied with the authors' responses and the resulting changes to the submitted work..............

Reviewer #3: accepted manuscript without any more comments, the authors addressed all questions clearly ,We sincerely appreciate your thorough review and valuable feedback on our

manuscript. While some of the points raised appear to relate to aspects not directly

addressed in our study (e.g., recall periods), we recognize the importance of

methodological clarity and have carefully reviewed all comments to ensure our

responses align with the study’s scope. Below are my detailed responses to the

comments. We hope these revisions meet your expectations. Should any concerns

remain, we are fully committed to further refining the manuscript. Thank you for your

time and expertise in improving our work.

Q:1- what is the justification for selecting this number of community reporters?

A:We appreciate the reviewer’s insightful question regarding the sample size

calculation in our study.

The sample size was calculated based on the primary outcome (respiratory

depression), assuming a 30% incidence in the remimazolam-propofol (RP) group and

a 50% relative reduction with remimazolam-etomidate (RE). With 80% power and a 5%

significance level, 118 patients per group were required. Accounting for a 5% dropout

rate, we enrolled 124 patients per group (total 248). This aligns with similar sedation

studies and ensures robust detection of clinically meaningful differences.

We hope this clarification addresses the reviewer’s concern.

Q:2- bias might be there if you have recruited members form local company especially

if they know about it before?

A:We appreciate the reviewer’s concern regarding potential bias in participant

recruitment.

We would like to clarify that all participants in this study were patients routinely

scheduled for gastrointestinal endoscopy at our hospital, with no prior knowledge of

their group assignment. The randomization process was computer-generated and

strictly implemented to ensure unbiased allocation to either the remimazolametomidate or remimazolam-propofol groups. Importantly, the study employed a doubleblind design where both participants and clinicians were unaware of group

assignments throughout the trial period.

We believe these measures effectively address the reviewer's concern regarding

recruitment and allocation bias in our trial.

Q:3- you can’t compare six months recall period by community informants vs 12

months from WHO

A:Thank you for your valuable feedback.

Our study is a prospective randomized controlled trial, with all data (e.g., respiratory

depression, hypotension) collected in real-time during the procedure and immediate

post-op period (≤24h). No retrospective or long-term follow-up data were included.

We are happy to revise the manuscript to address any specific concerns. Please let us

know if further clarification is needed.

Q:4- clarify hoe duplicated reports are handled

A:Thank you for your valuable feedback regarding the handling of duplicated reports in

our study. We appreciate your attention to methodological rigor. To clarify our

approach:

Prevention during enrollment: each participant was assigned a unique study ID.

Data collection and management: case report forms included signature fields by

anesthesiologists to verify procedural uniqueness.

Statistical analysis: no duplicates were detected in the final dataset.

If the above response fails to address your query, it may be due to our incomplete

understanding of your question. We kindly encourage you to provide more detailed

information, as we are committed to refining the academic quality of our content based

on your feedback.

Q:5- don't used under estimation in page 12 without any strong validation and bench

marking

A:Thank you for your valuable feedback on our manuscript. We sincerely appreciate

the time and effort you have dedicated to reviewing our work. Regarding your comment

about "under estimation in page 12 without any strong validation and benchmarking,"

we acknowledge that this point was not entirely clear to us, as our manuscript does not

explicitly mention "under estimation" on page 12 or elsewhere.

If we have misinterpreted your comment, we would be grateful for further clarification

regarding the specific aspect of "under estimation" you are referring to. This will allow

us to provide a more targeted response or make the necessary revisions to the

7. PLOS authors have the option to publish the peer review history of their article (what does this mean?). If published, this will include your full peer review and any attached files.

Reviewer #1: No

Reviewer #2: No

Reviewer #3: **Yes: **Yousra Nomeir

---

## [Editor Report · Acceptance letter]

PONE-D-25-09878R1

PLOS ONE

Dear Dr. Wang,

I'm pleased to inform you that your manuscript has been deemed suitable for publication in PLOS ONE. Congratulations! Your manuscript is now being handed over to our production team.

Kind regards,

on behalf of

Dr. Chih-Wei Tseng

Academic Editor

PLOS ONE